# Shoot Development through Modified Transverse Thin Cell Layer (tTCL) Culture of *Phalaenopsis* Hybrid Protocorms

**Kuo-Chin Lo [1], Jualang Azlan Gansau [2], Chia-Hung Shih [1] and Chien-Yuan Kao [1,*]**

[1] Department of Horticulture, College of Bioresource, National ILan University, Yilan 26047, Taiwan; mulder_lo@hotmail.com (K.-C.L.); 711711sch@gmail.com (C.-H.S.)

[2] Faculty of Science and Natural Resources, Universiti Malaysia Sabah, Kota Kinabalu 88400, Malaysia; azlanajg@ums.edu.my

[*] Correspondence: cykao@niu.edu.tw

**Abstract:** This first-attempt study used microtome-based methods to generate a thin cell layer culture for the micropropagation of Phal. Hwafeng Redjewel × Phal. New Cinderella. Protocorms were embedded in various agarose concentrations (8–12%, $w/v$) and dried from 1 to 8 h before sectioning with a microtome. Optimal conditions for slicing sections of 100 to 300 μm were achieved when the protocorms were embedded at 10% ($w/v$) agarose and dried for 4 h under laminar flow, and the hardness of the agarose block under these conditions reached $641.8 \pm 9.5$ g·cm$^{-2}$. The sectioned protocorms that were cultured on an MS medium supplemented with 1.2 mg·L$^{-1}$ 6-benzylaminopurine and 0.1 mg·L$^{-1}$ α-naphthaleneacetic acid were capable of growth and differentiated through the neoformation of protocorm-like bodies (PLBs) and/or callus before subsequent regeneration into plantlets and development into healthy plants in a nursery environment. The results of this study demonstrate that microtome-based tTCL is a reliable and promising approach for mass propagation and possible virus-free propagation objectives for *Phalaenopsis*.

**Keywords:** *Phalaenopsis*; shoot development; protocorm; transverse thin cell layer (tTCL)

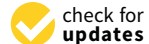



## 1. Introduction

Plant cells maintain a complex interaction with their neighbors during the differentiation of complex shapes, structures, and functions. However, this intrinsic information has been experimentally demonstrated as inhibitory in terms of in vitro morphogenetic potential [1]. To demonstrate this rationale of his hypothesis, Tran Thanh Van [2] developed and first described in tobacco culture a thin cell layer (TCL) technique employing small, excised explants from different plant organs.

Since the remarkable discovery of the TCL technique, various cellular, biochemical, and in vitro micropropagation aspects of several monocotyledonous and dicotyledonous plants have been examined with explants from roots, shoots, and somatic embryos to extend the potential in the embryogenesis/organogenesis of isolated plant tissues [3–11]. In all TCL-related papers published to date, the acquisition of thin cell layers has relied on freehand longitudinal or transverse slicing of the plant materials to generate small explants with inconsistent thicknesses generally greater than 0.5 mm. To improve this technology, the construction of a new TCL system is thus necessary.

Protocorms and protocorm-like bodies (PLBs) are tuberous embryonic masses of cells that are developed from seeds and vegetative tissues, respectively; they can grow into new plantlets and be applied in commercial micropropagation, such as in a bioreactor [12]. PLBs usually grow in cluster form due to their regeneration from sticking cells. Hence, protocorm culture is more practical for individual explants with homogenous forms and sizes for this study. NAA and BA are widely applied for the regeneration of shoots from protocorms or PLBs in many *Phalaenopsis* species or hybrids [13–15]. Agarose is a highly purified linear galactan hydrocolloid isolated from the *Gelidium* species of seaweed and is mostly used as a

gel matrix to separate, identify, and purify nucleic acids. Agarose is used as the embedding gel matrix material because of its properties of not being reliant on polymerization catalysts, a low gelling point, concentration-dependent rigidity, a highly visible gel matrix (easy for contamination assessment), easy and quick gel casting, a lower capability of air bubbles trapping in the matrix gel (and easier removal by vacuum degassing), and nontoxicity to cells for the following regeneration. Furthermore, this microtome-based method generates thin cell layers and induces rapid neoformation—an important factor in PLB microprop­agation and the virus elimination treatment of *Phalaenopsis* species. To our knowledge, this work provides the first report on agarose embedding, microtome-based tTCL, and the resulting regeneration into whole plantlets.

## 2. Materials and Methods

### 2.1. Surface Sterilization of the Phalaenopsis Capsule

A mature capsule of guided-pollinated *Phalaenopsis* hybrid (Phal. Hwafeng Redjewel × Phal. New Cinderella) was surface-sterilized with 70% (*v/v*) ethanol for 30 s followed by dipping in a 0.6% (*v/v*) solution of sodium hypochlorite containing 0.1% (*v/v*) Tween 20 (Sigma-Aldrich, St. Louis, MO, USA) for 10 min. The capsule was then rinsed three times with sterile distilled water and placed under a laminar flow hood for 10 min before being dissected longitudinally.

### 2.2. Asymbiotic Germination of Phalaenopsis Seeds

Asymbiotic germinations of *Phalaenopsis* seeds were performed on MS medium [16] supplemented with 2 g·L$^{-1}$ Bacto-tryptone (BD, Biosciences, Franklin Lakes, NJ, USA) and 35 g·L$^{-1}$ sucrose (Sigma Aldrich, St. Louis, MI, USA). The pH was adjusted to 5.4 before the addition of 10 g·L$^{-1}$ Bacto-agar (BD, Biosciences, Franklin Lakes, NJ, USA). Approximately 100 mL of medium was placed in a 500 mL conical flask and sterilized with an autoclave at 121 °C (15 psi) for 20 min. The seeds were spread on the surface of the sterile medium and incubated for 4 weeks in a controlled growth room at 26 ± 1 °C and a 16 h/8 h light/dark photoperiod with a light intensity of 30 mmol·m$^{-2}$·s$^{-1}$ provided by a Phillips® cool-white, fluorescent lamp. Visible protocorms of stage three [17] were observed at the end of the culture period.

### 2.3. Protocorm Embedding and Hardness Test

Mature protocorms (~3.5 mm diameter) of Phal. Hwafeng Redjewel × Phal. New Cinderella were air-dried in the laminar flow for 45 min before being used as the embedded material. The block mold box in normal paraffin section was used and the designated agarose solution was sterilized in an autoclave at 121 °C for 20 min. After sterilization, both were put in laminar flow chamber to maintain the sterile condition and cool down agarose solution temperature to 35–40 °C. The melted agarose solution was then poured into the block mold box and the protocorm was inserted into the agarose solution. A dissecting needle was used to clear the tiny bubble in the agarose solution. The resulting protocorm-embedded agarose block was then placed under the operating laminar flow hood for 4 h to reach a suitable hardness for microtome slicing. It was then sectioned at the desired thickness on the microtome and the tTCL section was cultured in the Petri dish with a sterile medium. Protocorms were embedded in 8, 10, and 12% (*w/v*) agarose (Agarose Type I-A: Low EEO, Sigma-Aldrich, St. Louis, MI, USA Product Number A0169, gelling point and melting point at 1.5% gel: 33 ± 1.5 °C and 87 ± 1.5 °C, respectively) and then air-dried under laminar flow for various durations (1 to 8 h) at room temperature (25 ± 1.0 °C) to determine the optimal rigidity for microtome sectioning. The percentage weight loss was calculated as the percentage of the weight difference between the initial and final dried weights of the agarose block over the drying duration. The hardness of the agarose block was analyzed using a texture analyzer (SMS TA-XT2, Godalming, UK). The success rate of slicing in different agarose concentrations was also determined through tTCL sliced at 100 μm by using an F-A22 Microtome Blade (extra thin) mounted on a Large Rotary Microtome (Tominga, Taipie, Taiwan).

### 2.4. tTCL Slicing and Culturing

The microtome sets were placed into laminar flow chamber (cabinet) and disinfected by wiping them with 75% (*v/v*) alcohol and then exposing them to ultraviolet lights for 30 min. Transverse thin cell layers (tTCL) of 100–500 μm were excised from the tip to the base (Figure 1) of the protocorm embedded in 10% (*w/v*) agarose (Sigma Aldrich, St. Louis, MI, USA). The tTCL explants were placed in Petri dishes (10 cm in diameter) containing 20 mL of agar (Difco, Sparks, MD, USA)-solidified MS medium [16] supplemented with 0.1–2.0 mg·L$^{-1}$ of separately α-naphthalene acetic acid (NAA) and 6-benzylaminopurine (BA) for plant regeneration. The cultures were incubated for 45 days in a controlled growth room at 26 ± 1 °C with a 16 h/8 h light/dark photoperiod at a light intensity of 100 mmol·m$^{-2}$·s$^{-1}$, as provided by a Phillips® cool-white, fluorescent lamp. The survival and contamination percentage were observed after 10 days, while the percentage of explants that induced PLBs and the number of induced PLBs per explant were determined at the end of cultivation. The survival rate was determined with the following equation: Number of sections protruding callus or PLB/Total number of sections sliced from embedded protocorm.

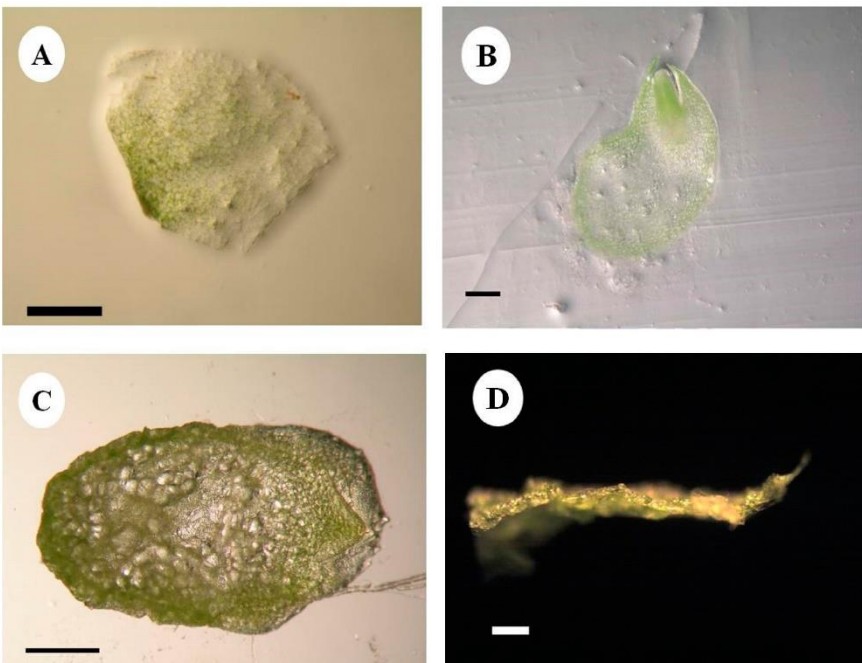

**Figure 1.** Different tTCL thicknesses of PLB sections generated by microtome: (**A**) 100 μm; (**B**) 200 μm; and (**C**) 300 μm. (**D**) is the side view of the 100 μm section as measured using a 0.01 m calibration plate under a microphotography system (bar = 100 μm).

### 2.5. Light Microscopy Analysis

For the microscopic analysis of protocorm and morphogenetic tTCL, the fresh tissue samples were fixed in FAA ((5% (*v/v*) formalin: 5% (*v/v*) acetic acid in 70% ethyl alcohol) for four hours and then dehydrated with a graded series of tertiary-butyl alcohol (TBA). Infiltration of the samples was carried out by gradual addition of paraffin wax (melting point 58–60 °C) until TBA solution attained supersaturation. The samples were then successively embedded into paraffin blocks and then sectioned with the aid of a Rotary Microtome (Tominga, Taipie, Taiwan) at 10–15 μm thickness. Dewaxing of the sections was performed by treating the specimen slides sequentially with xylol, xylol + alcohol, alcohol, water, and finally, with safranin staining [18], and the prepared sample slides were observed directly under a light microscope.

### 2.6. Shoot Development and Acclimatization

Shoots from the NAA and BA treated samples were subcultured on new MS medium [16] containing 20% (*w/v*) sucrose, 0.4 mgL$^{-1}$ BA, and 20% (*v/v*) green coconut water for further plant development. After 4 months, the well-developed plantlets with 3–4 leaves and 4–5 roots were isolated from the culture media and rinsed under running tap water before further treatment with 0.5% (*w/v*) Imas-Thiram 80 solution for 15 min. Plantlets were planted in pots containing moss and charcoal in a ratio of 1:1. The space on the upper part of the pot was covered by plastic with holes, and the pots were placed in a controlled environment with dim-light conditions and a temperature of $25 \pm 2$ °C.

### 2.7. Experimental Design and Data Analysis

Experiments were carried out in a randomized design and repeated twice, with each treatment having ten replicates. The data were subjected to two-way analysis of variant (ANOVA), and the mean values were compared by Duncan's multiple range test at $p < 0.05$ using the SAS program (SAS Institute, Cary, NC, USA).

## 3. Results and Discussion

### 3.1. Agarose as an Embedding Material for Protocorm tTCL Culture

Agarose solutions of different concentrations were employed for embedding to determine the optimal rigidity for microtome sectioning, followed by air-drying under laminar flow for various durations (Figure 2). The results indicated that total water loss (%) was not significant between agarose concentrations, but this gradually increased from 15 to 23%, with a drying time from 1 to 8 h. As expected, the 12% (*w/v*) agarose gel matrix was harder than the 10% (*w/v*) one, followed by the 8% (*w/v*) one. The hardness of the gel matrix increased gradually from $335.4 \pm 24.1$ to $602.2 \pm 25.8$ g·cm$^{-2}$, $504.8 \pm 17.9$ to $961.2 \pm 108.7$ g·cm$^{-2}$, and $675.3 \pm 8.4$ to $1617.4 \pm 402.2$ g·cm$^{-2}$ for 8, 10, and 12% (*w/v*) agarose, respectively, with the increment of drying time from 1 to 8 h.

The success of the slicing rate was further measured using 100 μm tTCL protocorm embedded in 8%, 10%, and 12% (*w/v*) of the agarose gel matrix, as shown in Table 1. The results indicated that when the concentration of the agarose solution was below 10% (*w/v*), the resulting high-water content (Figure 1A) prevented protocorm slicing and required a longer duration for agarose drying. The optimal (~80% success) agarose concentration for protocorm slicing was 10% (*w/v*), at which the hardness of the agarose block reached $641.8 \pm 9.5$ g·cm$^{-2}$ after drying for 4 h (Table 1). Under these conditions, protocorm sections of 100 μm could be easily sliced with homogeneous thickness. Although a shorter duration is required for drying, a higher concentration of agarose has a lower success rate of tTCL slicing.

Figure 3 shows the survival and contamination rates of protocorm tTCL explants embedded in a 10% (*w/v*) agarose gel matrix and dried for 4 h. The result indicates that the explants' survival rate decreased as the TCL thickness declined from 500 μm to 100 μm. Although the TCL system is designed to inoculate as small a number of cells as possible, the trait of being thin is not defined and measured by any methods, and 500 μm is usually claimed as the lower limit for the conventional freehand slicing of TCL. In the present study, the survival rates were not significantly different when sliced from 200 to 500 μm. Meanwhile, 100 μm TCL showed a dramatic drop to the lowest survival percentage ($15.0 \pm 6.2$%) among the thicknesses tested. Hence, 200 μm could be the ideal size for the slicing of Phal. Hwafeng Redjewel × Phal. New Cinderella protocorms. However, tTCLs as thin as 28 μm were successfully generated and survived in our laboratory, which could be an important potential for virus elimination treatment in Phalaenopsis. In addition, the contamination rate of protocorm tTCL was low and not affected by the tissue thickness.

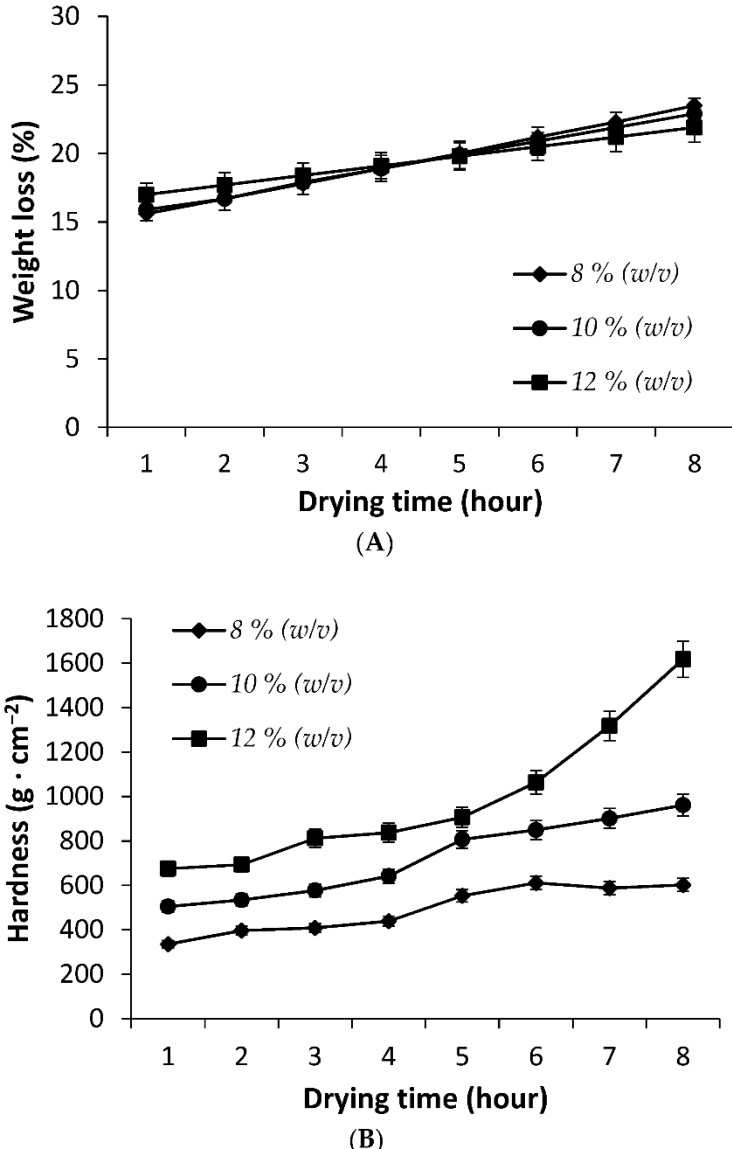

**Figure 2.** Effects of agarose concentration and drying time on the weight loss (**A**) and hardness (**B**) of the agarose block encapsulating the protocorm.

**Table 1.** Effect of agarose concentrations on the success rate of tTCL sections at 100 μm.

| Time (hour) | 8% | 10% | 12% |
|---|---|---|---|
| 1 | - | - | - |
| 2 | - | - | - |
| 3 | - | - | + |
| 4 | - | ++++ | ++ |
| 5 | + | ++ | + |
| 6 | ++ | + | - |
| 7 | + | - | - |
| 8 | - | - | - |

Note: -: 0%; +: 20%; ++: 40%; ++++: 80%.

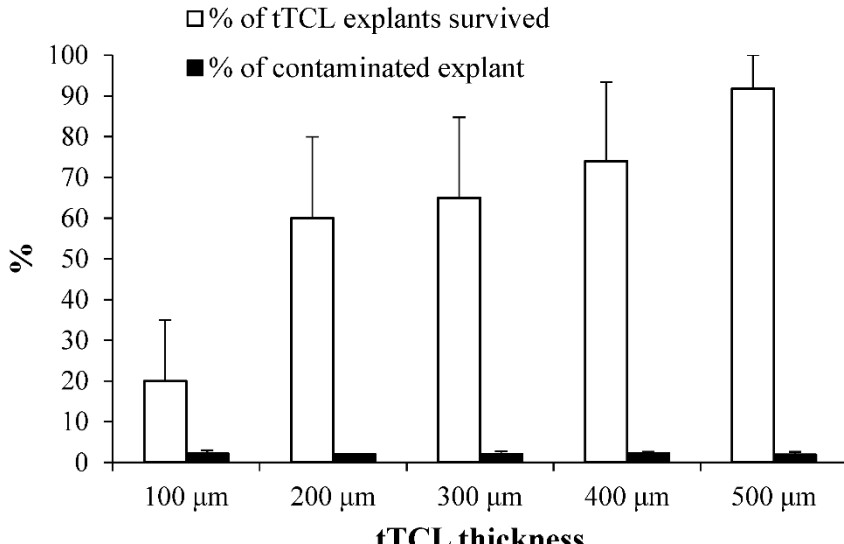

**Figure 3.** Effect of tTCL thickness prepared from 10% (*w*/*v*) agarose and dried for 4 h on survival and contamination rate of the cultures after 10 days.

In a preliminary test, agar, agarose, and phytagel were tested for their ability as the ideal embedding agent for the microtome section. The results (data not shown) indicated agarose to be the best among them in terms of gelling property, successful microtome slicing rate, and the section's survival rate. The nontoxic nature of agarose provided an advantageous application in the plant cell, tissue, and organ cultures. The main function of agarose in plant cell culture is to keep the cells from rupturing or aggregating, and also to stabilize membranes by inhibiting lipid peroxidation and preventing the leakage of cell wall precursors and other metabolites [19,20]. Lopez-Arellano et al. [20] reported the protoplast culture of *Stevia rebaudiana* by using a 1.2% (*w*/*v*) agarose bead or a thin layer liquid culture. In this technique, the protoplast culture successfully improved the division frequency by almost 1.5 times and showed a plating efficiency of 13% and 9.1%, respectively, with a survival rate of 23.5% to 14.8% compared to normal liquid culture. Callus induction and shoot regeneration were also observed in the protoplast culture of *Ulmus minor* [21], *Ulmus americana* [22], and *Gentiana macrophylla* [23] using 1.5–1.6% (*w*/*v*) of low-melting agarose beads.

### 3.2. Neoformation of Callus and PLB from Protocorm tTCL Culture

In this study, two pathways led to plant regeneration from the sliced tTCLs—protocorm-like bodies (PLBs) and calli. PLB neoformation was observed from the cutting surfaces of sections between 100 to 500 μm (Figure 4), and these PLBs could develop into complete plantlets. However, tTCLs as thin as 28 μm were successfully generated and survived in our laboratory, which could be an important potential for virus-elimination treatment in Phalaenopsis. Another pathway of regeneration was callus formation. Small, yellow, and granular calli usually result from the cutting surfaces of the sections. These calli easily formed PLBs without transferring to a different medium, suggesting that an embryogenesis callus with regeneration potential has been induced in these microtome-based sections. The PLBs and callus clump eventually developed into complete plantlets (Figure 4). These results also indicated that regeneration from a thin cell layer by callus and/or PLB induction mainly occurs on the protocorm epidermis surfaces.

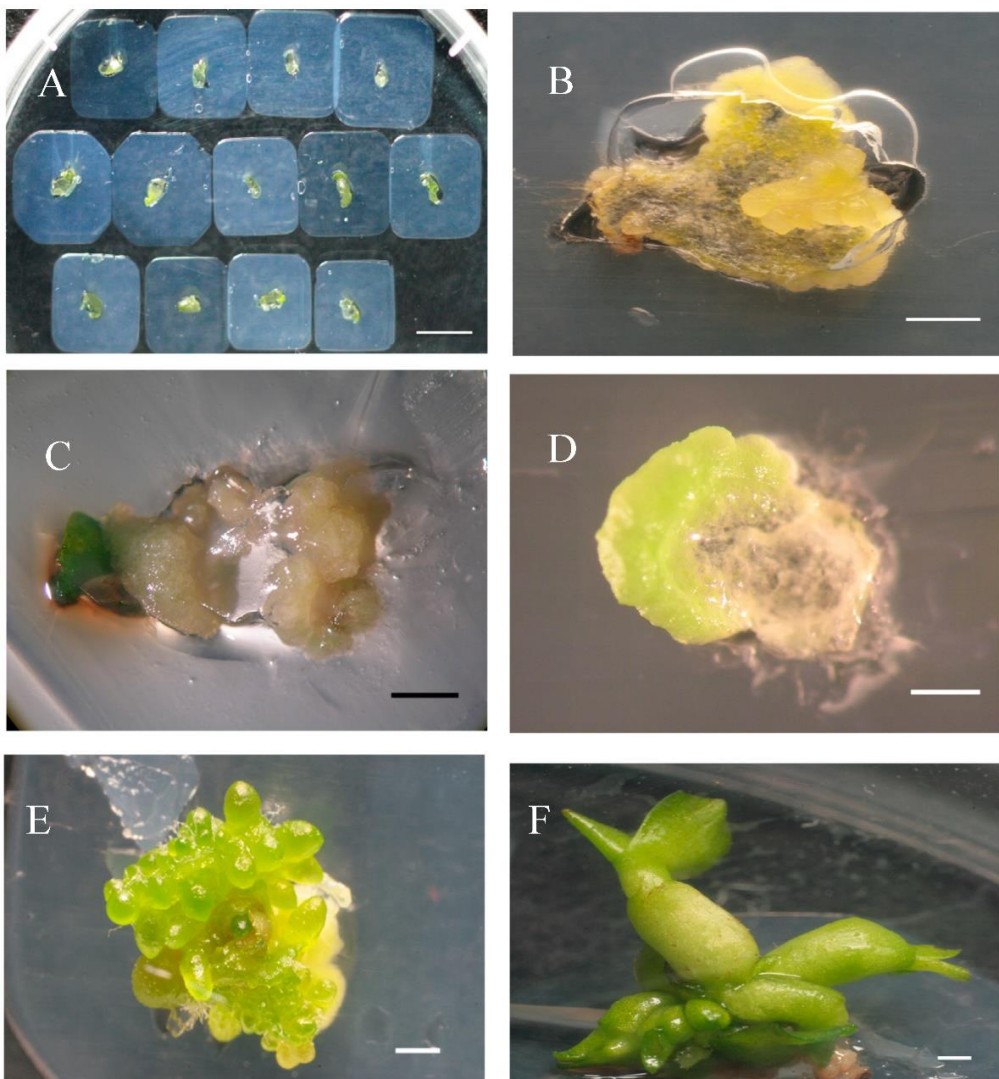

**Figure 4.** Two types of regeneration patterns from tTCL sections. (**A**) Serial arrangement of agarose tTCL sections generated by microtome slicing of PLB at 200μm (bar = 0.1 cm); (**B**) PLB neoformation in various development stages around the surface ring of the embedded tTCL (bar = 0.1 cm); (**C**) pale granular calli from the surface ring of the embedded tTCL (bar = 0.1 cm); (**D**) multiple cells occurring from the tTCL surface, which then developed into shoots (bar = 0.1 cm); (**E**) multiple PLBs regenerated from the outer surface of tTCL (bar = 0.5 cm); (**F**) after culturing for 90 days, the development of multiple shoots could be observed (bar = 0.1 cm).

　　Microscopic analysis on PLB induction revealed that PLBs and calli could be directly developed from the epidermis layers near the outer surface region (Figure 5). Similar findings were reported for other orchid species, such as *Doritaenopsis* hybrid [24] and *Dendrobium candidum* [25]. The histological observations reported here provide evidence of the morphogenetic pathways of the *Phalaenopsis* hybrid. Morphological changes were visible after 3 weeks of culture, while histological sections displayed cellular modifications as early as 5 days post-culture. The pattern of morphogenetic development reported here was produced by the simultaneous divisions of several cells and thus was multicellular in origin. This phenomenon may explain why the epidermis layer yields the most vigorous regeneration from protocorm slices [9,26]. Previous studies have reported similar results, showing that the TCL technique owes its success to the optimal induction of several centers of meristematic activity, which are generally present in the protocorms; however, the regeneration of a single shoot from each protocorm is also observed [9,27–29].

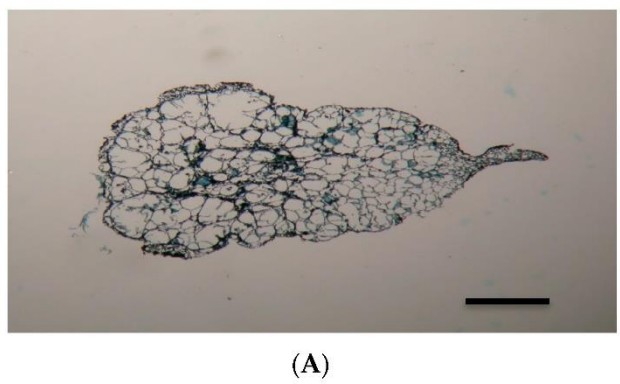
**(A)**

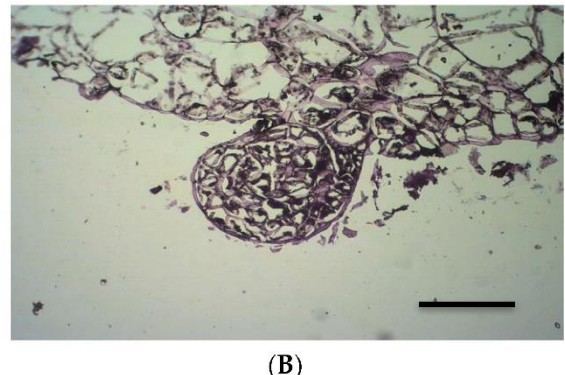
**(B)**

**Figure 5.** tTCL sections of *Phalaenopsis* hybrid protocorm showing PLB formation. (**A**) Initial section of tTCL (bar = 0.1 cm); (**B**) a developing globular-shaped PLB after 2–3 weeks of culture (bar = 0.02 mm).

### 3.3. Effect of PGRs on PLBs Formation

The effect of BAP and NAA at different concentrations on the induction of PLBs was assessed on *Phalaenopsis* hybrid tTCLs cultured on the MS medium (Table 2). Generally, the percentage of the explant response to PLB formation is enhanced by an increasing protocorm tTCL thickness. BAP treatment is also significantly better for PLB induction compared with NAA. However, the number of PLBs produced did not significantly differ between 200 and 500 μm tTCL, regardless of the types and concentrations of PGRs, while a lower number of PLBs was observed in 100 μm tTCL for both NAA and BAP treatments. The highest mean values of PLBs was 11.33 per explant, followed by 9.22 per explant, with a survival percentage of 93.11% and 95.22% observed on the medium supplemented with 1.2 mg·L$^{-1}$ BAP and 0.4 mg·L$^{-1}$ BAP, for 400 μm and 500 μm tTCL slices, respectively. Many orchid species require auxins and/or cytokinins for the neoformation of protocorm-like bodies (PLBs) and plantlet development. However, the ratio of auxin and cytokinin for PLB formation depends upon the species studied [30]. NAA was found to be more superior than other PGRs for the PLBs' induction, such as *Aranda* [31], *Dendrobium candidum* [25], and *Dendrobium aqueum* [32]. NAA is also frequently used in combination with cytokinins, similar to BAP in other orchid species [17,33,34]. However, our results demonstrated a better regeneration using BAP, although the differences between NAA and BAP were insignificant.

**Table 2.** Effect of PGRs and tTCL thickness on induction percentage and number of PLBs per explant in MS medium as observed after 45 days.

| | Thickness of tTCL Explant | | | | | | | | |
|---|---|---|---|---|---|---|---|---|---|
| | 100 μm | | 200 μm | | 300 μm | | 400 μm | | 500 μm | |
| PGRs (mg·L$^{-1}$) | % | No. | % | No. | % | No. | % | No. | % | No. |
| | PLBs Forming | | | | | | | | | |
| Control | 4.88 ef | 1.22 b | 29.44 b | 6.44 abc | 32.55 d | 4.88 bcd | 52.66 e | 4.00 def | 96.55 a | 5.11 bcd |
| NAA | | | | | | | | | | |
| 0.1 | 7.66 c | 3.44 a | 30.55 b | 4.44 bcde | 55.55 c | 3.88 def | 78.77 bc | 5.33 cd | 93.22 abc | 3.11 d |
| 0.5 | 5.77 de | 1.80 b | 25.88 bc | 2.66 e | 33.66 d | 2.11 ef | 67.66 d | 3.77 def | 94.66 ab | 7.33 abc |
| 1 | 5.22 def | 2.44 ab | 19.88 cd | 3.11 de | 23.66 e | 1.11 f | 51.77 e | 2.22 ef | 92.77 abc | 2.88 d |
| 1.5 | 4.20 f | 2.00 b | 10.22 e | 2.22 e | 20.88 e | 1.66 ef | 28.44 f | 1.77 f | 73.11 d | 3.11 d |
| 2 | 2.60 g | 1.20 b | 4.44 e | 2.00 e | 9.88 f | 1.88 ef | 14.22 g | 1.55 f | 56.00 e | 2.77 d |
| BA | | | | | | | | | | |
| 0.1 | 4.66 f | 1.33 b | 16.88 d | 7.55 a | 69.11 a | 7.11 abc | 84.55 b | 5.11 cde | 95.66 a | 3.88 cd |
| 0.4 | 6.00 d | 1.61 b | 30.66 b | 7.11 ab | 66.77 ab | 7.77 a | 82.77 b | 8.66 b | 95.22 ab | 9.22 a |
| 0.8 | 7.66 c | 2.44 ab | 26.66 bc | 5.77 abcd | 54.33 c | 7.66 ab | 71.33 cd | 6.00 bcd | 93.44 abc | 7.88 ab |
| 1.2 | 9.77 a | 3.44 a | 45.33 a | 4.00 cde | 65.44 ab | 7.55 ab | 93.11 a | 11.33 a | 94.55 ab | 4.33 cd |
| 1.6 | 8.66 b | 2.44 ab | 28.33 b | 3.11 de | 60.11 bc | 6.00 abcd | 70.11 cd | 7.11 bc | 88.77 c | 3.55 d |
| 2 | 5.11 def | 2.33 ab | 32.66 b | 2.88 de | 51.66 c | 4.44 cde | 76.33 bcd | 4.44 cdef | 90.44 bc | 7.33 abc |

Note: Mean value obtained from ten replicates. Data in the same column followed by the same letters are not significantly different at $p < 0.05$, using Duncan's multiple range test. NAA—α-naphthalene acetic acid; BA—6-benzylaminopurine.

Based on the study reported here, it can be concluded from the control treatments of Table 2 that thin protocorm sections of the Phalaenopsis hybrid produced PLBs in greater numbers in the 200 μm sample. Hence, for the optimal utilization of selected hybrid traits with significant explants' survival, a slice of 200 μm for the tTCL explants could be the applicable size in PLB production for subsequent shoot regeneration, and this concept can be exploited as a method of rapid plant propagation for Phal. Hwafeng Redjewel × Phal. New Cinderella. Furthermore, the histological study on the developmental pattern of PLB induction has led to a better understanding of the development of PLBs and their cellular origin.

## 4. Conclusions

In the present study, a transverse thin cell layer (tTCL) culture system was developed based on the microtome slicing of Phal. Hwafeng Redjewel × Phal. New Cinderella protocorm embedded in agarose. Protocorm slices in the range of 100 to 500 μm thickness were produced and subsequently regenerated into intact plantlets on MS medium [16] supplemented with various concentrations of α-naphthalene acetic acid (NAA) and 6-benzylaminopurine (BA). This paper is the first report of the application of a transverse thin cell layer (tTCL) culture in *Phalaenopsis* protocorms embedded in an agarose gel matrix. Protocorm embedded in 10% ($w/v$) agarose gel matrix, dried for 4 h, and sliced in 200 μm tTCL could be an optimum condition for Phal. Hwafeng Redjewel × Phal. New Cinderella. Under these conditions, protocorm sections of 100 μm could easily be sliced with a homogeneous thickness, as shown in this report. Furthermore, a longitudinal performance of the resultant tTCL to generate a thread or even a thin cell dot culture would have great implications in both application and academic areas for the Phalaenopsis orchid.

**Author Contributions:** Conceptualization, C.-Y.K.; methodology, C.-Y.K. and K.-C.L.; software, C.-H.S.; investigation, K.-C.L.; writing—original draft preparation, J.A.G. and C.-Y.K. All authors have read and agreed to the published version of the manuscript.

**Funding:** This research was funded by Council of Agriculture, Taiwan, ROC. Grant number [99AS-1.1.1-FD-Z2].

**Institutional Review Board Statement:** Not applicable.

**Informed Consent Statement:** Not applicable.

**Data Availability Statement:** Not applicable.

**Conflicts of Interest:** The authors declare no conflict of interest.

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
