# Peer review of "Shoot Development through Modified Transverse Thin Cell Layer (tTCL) Culture of Phalaenopsis Hybrid Protocorms"

_horticulturae, doi:10.3390/horticulturae8030206_

Round 1

Reviewer 1 Report

The paper „Plantlet Regeneration Through Modified Transverse Thin Cell Layer (tTCL) Culture of Phalaenopsis Hybrid Protocorms“ is focused on development of micropropagation protocol of orchid Phal. Hwafeng Redjewel × Phal. New Cinderella by use of thin cell layer culture. Authors evaluated influence of agarose concentration for protocorm embedding and time of agarose block drying to get optimal conditions for section slicing in the range of 100 to 500 μm.
The sectioned protocorms were cultured on MS medium supplemented with different concentrations of NAA and BAP. The regeneration of protocorm-like bodies (PLBs) and/or callus was observed and survival rate was evaluated.

The main contribution of this paper is that it represents the first study used microtome-based methods to generate a thin cell layer culture for the micropropagation of Phal. Hwafeng Redjewel × Phal. New Cinderella.

Although, the paper is well organized and experiments are properly established, I have some comments to the manuscript which are written in the attached „manuscrip with comments“. I see the main insufficiency in discussion and comparison of  the obtained results with the results of other authors. In general, in all manuscript authors mention the results of other authors just by designation of paper number without any extra detailed information, so the real comparison of results is not possible. Therefore, the discussion should be modified. Also some statements in obtained results have to be explained (see comments in manuscript).

I  do not feel competent to evaluate the English of manuscript.

  Conclusion: The paper can be published after minor revision according to the comments.

Reviewer 2 Report

Dear authors, I strongly advise you to have a native speaker to check the language, some parts are confusing to read and lack details.

Abstract and Introduction: When you write about cell to cell communication in your introduction, please use the right terms. "intertwined relationships" and "network of correlation between organs..."is incorrect. Your work is on protocorms which you introduce at the end of the introduction, it should be placed higher up, e.g. before you summarize your paper. "devirus" might be a lab term but is likely not right for the abstract.

M&M: You stress in your paper the use of agarose and that it has been use for embedding of e.g. protoplasts. The agarose use for protoplasts is a very specific one and from your M&M it is not clear which type you used. I further miss which type of mold you used for embedding, how you mount the agaose block into your microtome. Why were hand-sections not taken as a control e.g. if these data can't be found in the literature?

Your title says : "Plantlet regeneration" but the regeneration is not described, only shoot development.

How many sections where placed per petridish? Are the "ten replicates" then different petridishes with a number of sections each?

R&D: Please define "success of slice rate". Is this base on all sections (slices) you took, % of damaged or incomplete sections?

I can't follow your conclusion that sections of 200 um are ideal, according to Figure 3 the survival rate of the 500 um one is much higher. Please explain.

Are the microtome section better than the free-hand sections (from the literature) with respect to survival rate and yield. Why should one use the more laborious microtome technique when free-hand sections are easier and most likely quicker?

Line 211: "Neoformation"  .. of what?

Line 213: "two main pathways" where there other (minor) pathways as well?

Further: Do the PLBs derive from the same cell layer as the callus? Do you see different types of callus ? What comes first callus or PLBs?

Line 228: What are the "morphological changes" ?

Figure 5: please indicate in picture A from where picture B was taken

Line 259: There is not table named  "Table 2" The table you provide in this section lacks title and legend. It is unclear how the data were generated, thus from the 10 replicates or individuals sections/treatment. What does "a, b,c,d,e,f" mean?

Author Response

Point 1: Dear authors, I strongly advise you to have a native speaker to check the language, some parts are confusing to read and lack details.

Response 1: The revised manuscript will be sent to English editing provided by MDPI.

Point 2: Abstract and Introduction: When you write about cell to cell communication in your introduction, please use the right terms. "intertwined relationships" and "network of correlation between organs..."is incorrect.

Response 2: "intertwined relationships" and "network of correlation between organs..." are revised as “ a complex interaction” and “intrinsic information” respectively.

Point 3: Your work is on protocorms which you introduce at the end of the introduction, it should be placed higher up, e.g. before you summarize your paper.  

Response 3: The protocorm’s description is replaced up in the paragraph as suggested

Point 4: "devirus" might be a lab term but is likely not right for the abstract.

Response 4: Revised as “virus free propagation”.

Point 5: M&M: You stress in your paper the use of agarose and that it has been use for embedding of e.g. protoplasts. The agarose use for protoplasts is a very specific one and from your M&M it is not clear which type you used.

Response 5: Agarose Type I-A: Low EEO , Sigma Product Number A0169

Point 6: I further miss which type of mold you used for embedding, how you mount the agaose block into your microtome.

Response 6: The block mold box as in normal paraffin section used and the designated agarose solution were sterilized in an autoclave at 121°C for 20 min. After sterilization, both are put in laminar flow chamber to maintain the sterile condition and cool down agarose solution temperature to 35-40°C. Pure the melted agarose solution into block mold box and then insert the protocorm, and use dissecting needle to clear the tiny bubble. The resulted protocorm embedded agarose block mold is then placed under the operating laminar flow for 4 hours for its suitable hardness for microtome slicing. Section the protocorm embedded agarose block at the desired thickness on the microtome and then culture the resulting tTCL in the petridish with medium.

Point 7: Why were hand-sections not taken as a control e.g. if these data can't be found in the literature?

Response 7: In all TCL-related papers published to date, the acquisition of thin cell layers has relied on freehand slicing of the plant materials to generate small explants with inconsistent thickness of generally greater than 0.5 mm. Hence it is impossible to hand slice the protocorm to the designated thickness and use as the control.

Point 8: Your title says : "Plantlet regeneration" but the regeneration is not described, only shoot development.

Response 8: The title words "Plantlet regeneration" is changed as “Shoot development”.

Point 9: How many sections where placed per petridish? Are the "ten replicates" then different petridishes with a number of sections each?

Response 9: Around 20 tTCL explants of designated thickness were sliced from each individual embedded protocorm by Rotary microtome and cultured in the same petridish. Each treatment has 10 embedded protocorm as replication.

Point 10: R&D: Please define "success of slice rate". Is this base on all sections (slices) you took, % of damaged or incomplete sections?

Response 10: the success of slice rate was determined with the following equation: Number of sections protruding callus or PLB / Total number of sections sliced from one embedded protocorm.

Point 11: I can't follow your conclusion that sections of 200 um are ideal, according to Figure 3 the survival rate of the 500 um one is much higher. Please explain.

Response 11: It can be concluded from the control treatments of Table 2 that thin protocorm sections of Phalaenopsis hybrid produced PLBs in greater numbers in the 200 μm.

Point 12: Are the microtome section better than the free-hand sections (from the literature) with respect to survival rate and yield. Why should one use the more laborious microtome technique when free-hand sections are easier and most likely quicker?

Response 12: Free-hand slicing is not able to generate TCL as thin as reported in this manuscript. Microtome slicing induces callus proliferation which is not present in most cases of convectional free-hand sections of Phal. species, and microtome slicing also provides the possible virus elimination potential in the future.

Point 13: Line 211: "Neoformation"  .. of what?

Response 13: Neoformation of callus and PLBs

Point 14: Line 213: "two main pathways" where there other (minor) pathways as well?

Response 14: "two main pathways" is changed to "two pathways".

Point 15: Further: Do the PLBs derive from the same cell layer as the callus?

Response 15: We do not intend to notice this trait.

Point 16: Do you see different types of callus ?

Response 16: No

Point 17: What comes first callus or PLBs?

Response 17: Callus

Point 18: Line 228: What are the "morphological changes" ?

Response 18: "morphological changes" of enlargement or shrink of the explants could be resulted from cell growth or death.

Point 19: Figure 5: please indicate in picture A from where picture B was taken

Response 19: Fig 5A and B show the microscopy images of tTCL section and developing PLB from the tTCL section, these two photos are not related each other.

Point 20: Line 259: There is not table named  "Table 2" The table you provide in this section lacks title and legend. It is unclear how the data were generated, thus from the 10 replicates or individuals sections/treatment. What does "a, b,c,d,e,f" mean?

Response 20: “Table 2. Effect of PGRs and tTCL thickness on induction percentage and number of PLBs per explant in MS medium as observed after 45 days” is added as title and the meaning of "a, b,c,d,e,f" has added as well.

Round 2

Reviewer 2 Report

Dear authors, the manuscript has improved a lot. Thank you for following the suggestions.

I think on page 4 there is a change in letter type, likely due to all the inserted changes. Please check.

Good luck and success with your future research,

Best regards, a reviewer